# MechSci: Scaling Clinical Science via Mechanistic Interpretability of Medical Foundation Models

**MechSci**  **Robbie Holland**[1]  **Ashwin Kumar**[1]

**Eduardo Pontes Reis**[1]  **Akshay S. Chaudhari**[1]  **Sergios Gatidis**[1]

[1]Stanford Center for Artificial Intelligence in Medicine and Imaging, Stanford University

## Abstract

Large, multimodal medical datasets harbor complex, latent structures that hold immense potential for scientific discovery. While foundation models excel at extracting predictive signals from such data, their inherent opacity limits their use as tools for generating new scientific knowledge. This work introduces a fully automated pipeline to uncover novel scientific knowledge by transforming the latent representations of medical foundation models into sparse, human-interpretable concepts. We employ Matryoshka Top-K Sparse Autoencoders (SAEs) to decompose dense feature vectors from a 3D CT imaging foundation model into a sparse basis of learned concepts. An automated interpretation module then uses a large language model to assign a semantic, clinical description to each discovered concept. Finally, the system systematically evaluates each concept for its prognostic value across a range of clinical outcomes, generating testable hypotheses. This entire process, from concept discovery to the generation of this manuscript, is automated. As a proof-of-concept, we present a detailed analysis of one such automatically generated hypothesis: a novel imaging biomarker, `image_Concept_66`, which LLMs concluded to represent "abnormal soft tissue density and stranding in abdominal fat." This feature is shown to be a strong predictor for the future onset of skin cancer (`cancer_skin`), with an odds ratio of 3.7 ($p < 0.001$), significantly outperforming clinical risk factors such as patient age, sex, race and BMI and smoking status. This work demonstrates a scalable, end-to-end system that transforms AI from a predictive tool into an engine for generating interpretable and clinically valuable scientific hypotheses.

## 1  Introduction

The increasing availability of large-scale, multimodal healthcare datasets presents an unprecedented opportunity for scientific discovery. Foundation models, including large language models (LLMs) and vision-language models (VLMs), have proven adept at learning rich, predictive representations from this data, often achieving performance that meets or exceeds that of human experts on specific diagnostic and prognostic tasks (1; 2; 3). However, the clinical and scientific utility of these models is fundamentally limited by their "black box" nature. The complex, high-dimensional latent spaces learned by these models do not readily map to human-understandable concepts, making it difficult to understand *why* a model makes a particular prediction and, consequently, to derive new scientific insights from its learned patterns.

Prior approaches to model interpretability, such as gradient-based or attribution-based saliency methods, have shown limited utility. These methods typically highlight which input features were important for a single prediction but fail to reveal the abstract, high-level concepts the model has learned and uses consistently across the dataset. To bridge this gap and unlock the scientific potential

of foundation models, a move towards mechanistic interpretability is required. This paradigm seeks to deconstruct a model's internal computations into their constituent, understandable parts.

In this work, a novel framework is proposed to achieve mechanistic interpretability in the context of multimodal medical data. The core idea is to transform the opaque, dense feature spaces of foundation models into sparse and interpretable concept sets by training Sparse Autoencoders (SAEs) (4). SAEs are unsupervised models designed to learn an overcomplete dictionary of features, forcing any given input to be reconstructed from a sparse linear combination of these dictionary features. Recent work has shown that these sparse features often correspond to semantically meaningful and monosemantic concepts (4; 5). We hypothesize that applying this technique to features from medical foundation models can uncover clinically relevant concepts, linking model predictions to human-interpretable phenomena.

Our primary contribution is an end-to-end, automated pipeline that operationalizes this vision (Figure 1). The system begins by training Matryoshka Top-K SAEs (6; 7) on image embeddings from a medical foundation model. It then employs an LLM-based auto-interpretation module to assign clinical descriptions to the discovered sparse features. Finally, it systematically tests each feature for association with future clinical outcomes, generating a ranked list of novel, data-driven hypotheses. The scalability of this approach is demonstrated by its ability to generate thousands of such hypotheses automatically.

To showcase the pipeline's efficacy, this paper itself serves as an example of its output. The system identified a promising hypothesis, collated the relevant statistics and figures, and generated this manuscript for submission in a single shot (see A.1 for the single prompt). We present a detailed analysis of one such discovery: a novel imaging feature, `image_Concept_66`, that is highly predictive of future skin cancer diagnosis. This feature, interpreted by our system as representing "abnormal soft tissue density and stranding within mesenteric, peritoneal, or retroperitoneal fat," exhibits a significantly stronger association with skin cancer risk than established clinical risk factors. This finding not only highlights a potentially novel biological link between systemic inflammation and skin cancer but also validates our approach as a powerful engine for automated scientific discovery.

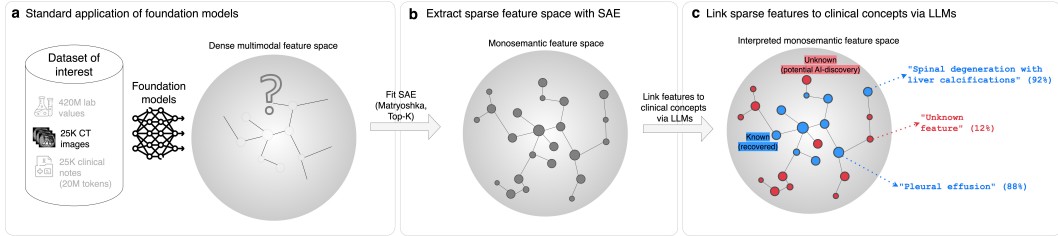

Figure 1: Synopsis of the automated discovery pipeline. (a) Foundation models are used to extract rich feature representations from large-scale, multimodal clinical data. (b) Matryoshka Top-K Sparse Autoencoders (SAEs) are trained on these features to decompose them into a sparse set of interpretable concepts. (c) A large language model-based auto-interpretation module assigns semantic labels to each concept and the system systematically tests for associations with clinical outcomes, generating novel scientific hypotheses.

## 2 Methods

The methodology is designed as a fully automated, multi-stage pipeline, from data processing to hypothesis generation and reporting.

### 2.1 Dataset and Cohort

The study was conducted using a large, de-identified, IRB-approved dataset from Stanford Health Care (SHC). This multimodal dataset includes 3D Computed Tomography (CT) images, associated radiology reports, laboratory values, clinical notes, ICD codes, and longitudinal outcomes. For this initial investigation, a cohort of 25,334 patients from Stanford SHC was used, which was split into train, validation and test using the Merlin splits (1). The system is designed for future generalization testing on a separate dataset from Stanford Health Care Tri-Valley.

## 2.2 Foundation Model Feature Extraction

The first stage of the pipeline involves the extraction of high-dimensional feature representations from the raw medical data. For the analysis presented in this paper, which focuses on imaging data, features were extracted from 3D CT scans using Merlin (2), a state-of-the-art vision-language foundation model for 3D medical imaging developed by our group. Merlin provides dense 1024-dimensional embeddings that capture rich radiological information from each CT scan. These embeddings serve as the input for the subsequent SAE training phase. The full pipeline is also equipped to process textual data (e.g., radiology reports, clinical notes) using LLMs such as Gemini or Qwen, though this modality is not the focus of the current case study.

## 2.3 Sparse Autoencoder Training

To transform the dense Merlin embeddings into an interpretable feature set, Sparse Autoencoders (SAEs) were trained. An SAE learns to represent an input vector $x$ as a sparse linear combination of an overcomplete set of dictionary vectors. All SAEs used in this work employ a Top-K activation function, where for each input, only the $K$ features with the highest activation are retained. The encoder $f(x)$ and decoder $\hat{x}$ are defined as:

$$f(x) = \sigma(\mathbf{W}_{\text{enc}}x + b_{\text{enc}}) \tag{1}$$

$$\hat{x} = \mathbf{W}_{\text{dec}}f(x) + b_{\text{dec}} \tag{2}$$

$$\mathcal{L}_{Standard}(x) = \|x - \hat{x}\|_2^2 \tag{3}$$

where $\sigma$ is the Top-K sparse activation function which sets all but the top $K$ values of its input to zero.

A key innovation in our approach is the use of Matryoshka Top-K SAEs (7). This architecture improves upon standard SAEs by learning a nested, hierarchical set of features. The dictionary vectors are ordered such that any prefix of the dictionary forms a valid, smaller dictionary. This is enforced by a modified loss function that penalizes reconstruction error at multiple dictionary sizes simultaneously:

$$\mathcal{L}_{Matryoshka}(x) = \sum_{m \in M} \|x - (\mathbf{W}_{\text{dec}}^{0:m} f(x)^{0:m} + b_{\text{dec}})\|_2^2 \tag{4}$$

where $M$ is a set of nested dictionary sizes. This structure allows for more efficient learning of the underlying concept space and provides features at varying levels of granularity. For this study, all Matryoshka Top-$K$ SAEs used a maximum dictionary size of $N = 8192$ and layer sizes $M = [128, 512, 2048, 8192]$. In total, we trained eight SAEs: four Matryoshka SAEs and four standard Top-$K$ SAEs, using four levels of sparsity $K \in \{5, 10, 20, 40\}$.

## 2.4 Automated Concept Interpretation

A critical component of the pipeline is the ability to automatically assign semantic meaning to the learned sparse features. Following the methodology of recent work in auto-interpretation (5), an LLM-based system was implemented. For each learned SAE feature, the 20 patient samples (CT scans) that produced the highest activation for that feature were identified. The anonymized "Findings" sections from the corresponding radiology reports for these 20 samples were then provided as context to a large language model (Gemini 2.5 Pro). The model was prompted to summarize the common radiological properties described across these reports into a concise, descriptive label.

To validate the generated interpretation, the label was then tested for its applicability on a held-out set of 20 different highly activating samples for the same feature. The percentage of held-out samples for which the interpretation was deemed accurate by the LLM serves as a generalization score for the proposed concept label.

## 2.5 Automated Hypothesis Generation and Reporting

The final stage of the pipeline is a large-scale, automated hypothesis generation engine. In a PheWAS (phenome wide association study), each of the thousands of interpreted SAE features was systematically tested for its association with a panel of 39 predefined clinical outcomes, with onset windows ranging from 0.5 to 7.5 years post-scan. This process generated a vast number of potential

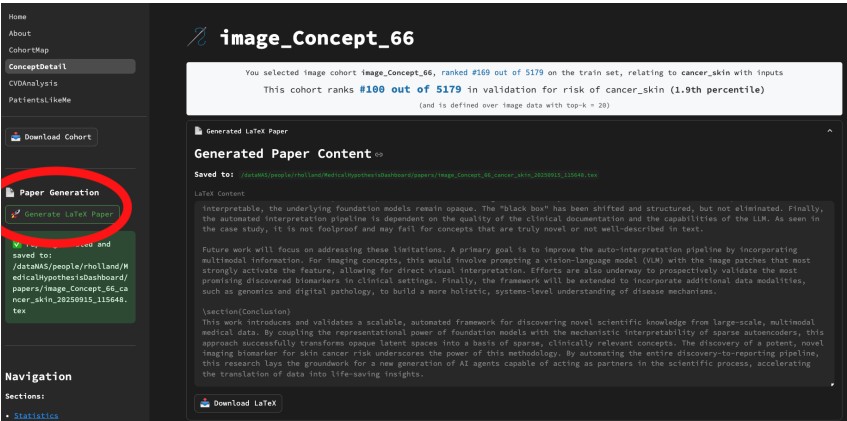

Figure 2: A screenshot of the "Feature Discoverer" web application. This interface allows for the exploration of discovered concepts and their association with clinical outcomes. The "Generate LaTeX paper" button, circled in red, initiates the automated manuscript generation process for the selected hypothesis and which created this paper.

scientific hypotheses, to which we apply multiple test correction. For the four SAE configurations, this resulted in:

- K=5: 6,357 hypotheses
- K=10: 6,942 hypotheses
- K=20: 7,605 hypotheses
- K=40: 13,143 hypotheses

This totals 34,047 unique feature-outcome pairs, each representing a potential discovery.

These hypotheses are presented to users via a web interface, the "Feature Discoverer" (Figure 2). From this interface, a user can select a promising hypothesis and trigger an agentic workflow by clicking a "Generate LaTeX paper" button. This action initiates a process where the system automatically gathers all relevant statistics, plots, and interpretations for the selected hypothesis and populates a predefined LaTeX template. The result is a complete, submission-ready manuscript, such as the one presented here, generated in a single shot with Gemini 2.5 Pro.

## 3 Results

The results demonstrate the success of the end-to-end pipeline, from the effective training of SAEs to the discovery and validation of a novel, clinically significant imaging biomarker.

### 3.1 Matryoshka SAEs Learn High-Fidelity and Performant Representations

The Matryoshka Top-K SAEs were found to effectively learn sparse representations of the CT image embeddings while maintaining high reconstruction fidelity. Figure 3a shows that Matryoshka SAEs (orange) consistently achieve a higher coefficient of determination ($R^2$) for a given number of alive features compared to standard Top-K SAEs (blue), demonstrating a superior Pareto frontier and more efficient learning. Furthermore, the learned sparse features retain the predictive power of the original dense embeddings. Figures 3b,c demonstrate that this trend extends to downstream clinical tasks.

### 3.2 Case Study: A Novel Imaging Biomarker for Skin Cancer

From the thousands of hypotheses generated, the system flagged the association between feature `image_Concept_66` (from the K=20, N=8192 SAE) and the future onset of `cancer_skin` as particularly strong. A detailed analysis of this hypothesis was conducted on a held-out test set.

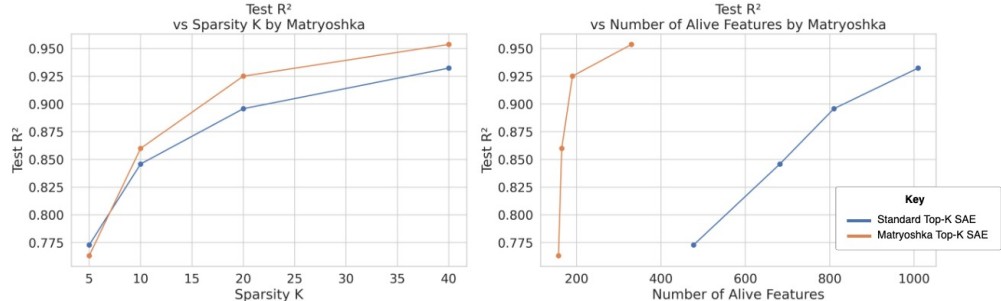

(a) Reconstruction fidelity by sparsity level K and the resulting number of alive SAE features

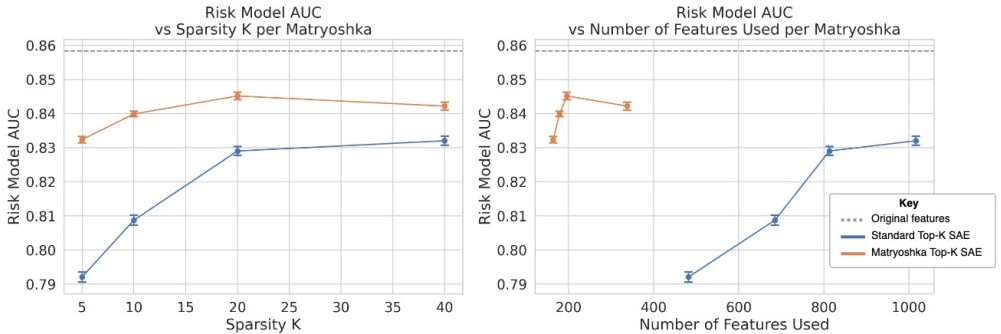

(b) Average performance over clinical downstream tasks for disease progression (95% confidence intervals calculated using over tasks using a t-distribution)

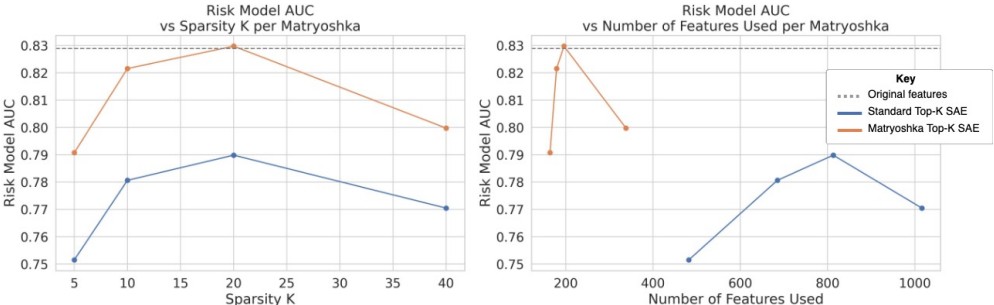

(c) Downstream task performance (progression to skin cancer within 0.5-7.5 years)

Figure 3: SAE Training and Downstream Task Performance with pareto improvement for Matryoshka SAEs. **(a)** Reconstruction fidelity ($R^2$) of the feature space versus the number of alive features for Matryoshka SAEs (orange) and standard Top-K SAEs (blue). **(b)** Downstream performance of logistic regression models trained on Matryoshka SAE features versus standard Top-K SAEs features averaged over a selection of diverse prognostic tasks for onset to blood cancer, abdominal cancer, skin cancer, dementia, diabetes and obesity using the 0.5 to 7.5-year window. We find the pareto improvement in feature space translates to downstream clinical tasks. **(c)** Downstream performance on skin cancer.

### 3.2.1 Automated Interpretation of `image_Concept_66`

The auto-interpretation module analyzed the radiology reports associated with high activations of `image_Concept_66`. All reports used in (2) have been approved for public use. The LLM synthesized the findings and concluded that the feature represents:

> "Abnormal soft tissue density and stranding within mesenteric, peritoneal, or retroperitoneal fat."

scoring 60% generalization and a 3/3 Anthropic AutoInterp rating. The LLM's reasoning noted that this visual pattern was a common denominator for various underlying pathologies, including post-surgical changes, inflammation (e.g., peritonitis, diverticulitis), fluid collections, and infiltrative masses, all of which disrupt normal anatomical fascial planes and appear as increased density in abdominal fat on CT scans. See A.2 for the full AutoInterp reasoning output.

### 3.2.2 Prognostic Value of `image_Concept_66`

The prognostic significance of `image_Concept_66` for predicting the onset of skin cancer within a 0.5 to 7.5-year window was evaluated. The feature demonstrated strong predictive performance, as detailed in Table 1. Notably, when analyzed in isolation, the presence of this feature was associated with an odds ratio (OR) of 3.7 ($p < 0.001$) for developing skin cancer. The feature was present in 7.7% of the test cohort. A clear dose-response relationship was observed, where higher activation values of the feature corresponded to a progressively higher incidence of skin cancer (Figure 4).

Table 1: Statistical Performance of `image_Concept_66` for Predicting `cancer_skin` on the held-out test set. Confidence intervals calculated using Wilson's test.

| Metric | Value |
|---|---|
| Feature Prevalence | 7.3% (64/877) |
| Baseline Incidence Rate of skin cancer | 33.52% (294/877) |
| Precision (binary feature) | 58.1% (95CI [51.4% - 64.4%]) |
| Specificity (binary feature) | 42.9% (95CI [37.3% - 48.6%]) |
| **Isolated Odds Ratio (per std.)** | **3.695** (95CI [2.840 - 4.808]) (p=0.0) |

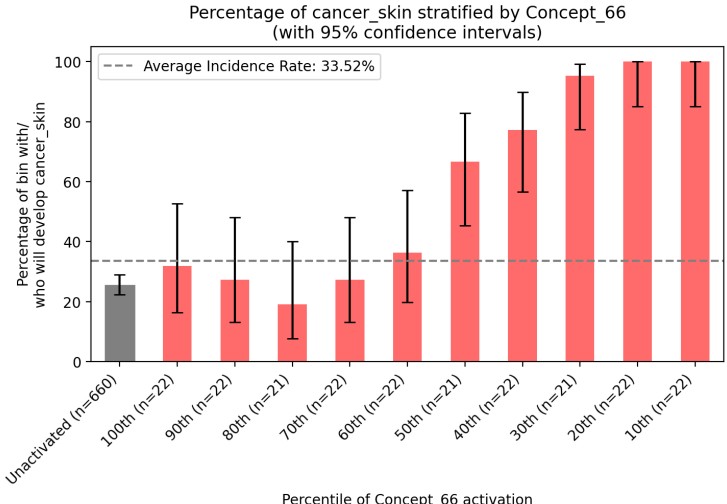

Figure 4: Dose-response relationship between the activation of `image_Concept_66` and the incidence of `cancer_skin`. The x-axis shows bins of feature activation values, while the y-axis shows the proportion of individuals in each bin who developed skin cancer.

### 3.2.3 Comparison with Clinical Risk Factors

The predictive power of `image_Concept_66` was compared with several widely available clinical risk factors for skin cancer, including routinely collected demographic and lifestyle data: patient age, sex, body mass index (BMI), race (White, Black, Asian, or other), and current smoking status. To ensure all SAE-derived and clinical features are comparable, we first independently normalize each so that odds ratios are in units of standard deviations.

As shown in Table 2, the discovered imaging feature has a substantially larger effect size than any of the conventional risk factors analyzed. A common clinical risk factor for skin cancer, patient age, had an odds ratio per standard deviation of 1.815. In contrast, the isolated odds ratio for the binary

presence of `image_Concept_66` was 3.695. Other factors such as smoking status were not found to be statistically significant predictors in this cohort.

Table 2: Comparison of odds ratios for `cancer_skin`. The discovered feature, `image_Concept_66`, demonstrates a significantly stronger association with disease onset than clinical, human-engineered risk factors. Confidence intervals for odds ratios calculated using Wald's method.

| Risk Factor | Odds Ratio per Std. Dev. (95% CI) | P-value |
|---|---|---|
| `image_Concept_66` | **3.695 (2.840 - 4.808)** | **< 0.001** |
| Race (White) | 2.007 (1.483 - 2.716) | < 0.001 |
| Patient Age | 1.815 (1.415 - 2.328) | < 0.001 |
| BMI | 1.623 (1.011 - 2.611) | 0.045 |
| Patient Sex (Male vs. Female) | 1.466 (0.981 - 2.191) | 0.062 |
| Current Smoker | 1.105 (0.851 - 1.435) | 0.455 |
| Race (Other) | 1.061 (0.701, 1.607) | 0.021 |
| Race (Black) | 0.706 (0.463, 1.076) | 0.106 |
| Race (Asian) | 0.611 (0.426, 0.877) | 0.007 |

### 3.3 Robustness of the Discovery Process

The discovery of potent predictive features was not unique to a single SAE configuration. Strong predictors for `cancer_skin` were identified across different SAEs trained with varying sparsity levels, indicating that the discovery process is robust. Figure 5 shows the odds ratio of the single best feature discovered for this outcome from each of the four trained SAEs, all of which show a strong predictive signal.

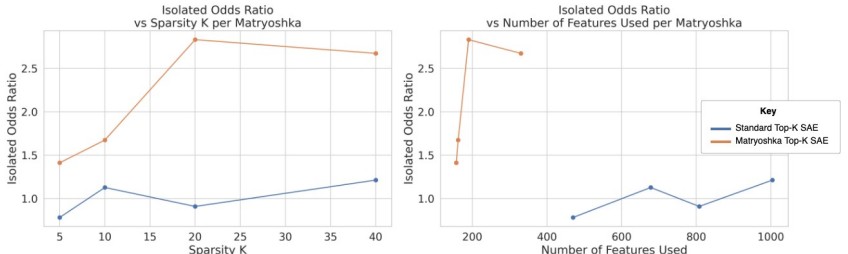

Figure 5: The highest odds ratio feature for `cancer_skin` discovered by Matryoshka SAEs (orange) and standard Top-K SAEs (blue). After ranking features, the plot reports the odds ratio on the held out test set (2). As before, we observe a pareto improvement using Matryoshka SAEs. Moreover, strong predictive features translated to test set, demonstrating the robustness of the approach.

## 4 Discussion

The results presented herein demonstrate the successful implementation of an automated pipeline for uncovering novel scientific knowledge from the latent spaces of medical foundation models. By leveraging Matryoshka Top-K SAEs and an LLM-based interpretation module, our system transforms opaque, high-dimensional embeddings into a vocabulary of sparse, interpretable clinical concepts. The subsequent automated screening of these concepts against clinical outcomes creates a scalable engine for hypothesis generation, culminating in the automated production of scientific manuscripts to report its findings.

The case study of `image_Concept_66` provides compelling evidence for the utility of this approach. The system independently discovered a novel imaging biomarker—"abnormal soft tissue density and stranding in abdominal fat"—and identified its strong association with the future development of skin cancer. The radiological sign of fat stranding is a non-specific indicator of inflammation. Its presence could reflect a state of chronic systemic inflammation, which has been previously hypothesized as a risk factor for various malignancies, including skin cancer. Our finding provides quantitative, image-based evidence for this link and suggests a new, potentially quantifiable risk factor that is not

captured by traditional measures. The fact that this automatically discovered feature outperforms clinical risk factors like age underscores the potential for this methodology to uncover signals that have been missed by conventional, human-driven research.

The primary strength of this work lies in its end-to-end automation and scalability. The system is capable of generating and evaluating tens of thousands of hypotheses without human intervention. This represents a paradigm shift from traditional medical research, which is often limited by the time and resources required to formulate and test each hypothesis individually. By automating this process, we can explore the vast combinatorial space of feature-outcome relationships, systematically searching for novel connections that can drive future clinical and biological investigation.

Despite these promising results, several limitations must be acknowledged. Crucially, the findings are, at this stage, correlational. While `image_Concept_66` is a strong predictor, this does not imply a causal relationship. The feature could be a marker for a confounding factor that is the true driver of risk. While AutoInterp concluded the feature was inflammation based, it is likely that this feature may detect evidence of metastatic or treated cancer that was missed or not recorded by the associated radiology report, or was not recorded in the patient's history at Stanford SHC. In this case, `image_Concept_66` would have use in flagging and stratifying cancer risk among incoming patients from different hospital systems.

Moreover, this analysis was conducted on data from a single academic medical center. The generalizability of the discovered concepts and their prognostic value must be validated on external datasets from different patient populations and healthcare systems. Our planned future work includes testing the generalization of discovered concepts on the SHC Tri-Valley dataset. Third, the LLM-based auto-interpretation, while powerful, is not infallible. The generated labels are hypotheses about the feature's meaning and require validation by clinical experts.

Future work will focus on addressing these limitations and expanding the scope of the pipeline. A key next step is to extend the SAE training to be truly multimodal, incorporating features from clinical text, laboratory values, and genomic data simultaneously. This will allow for the discovery of more complex, cross-modal concepts that may yield even deeper biological insights. Furthermore, we plan to establish a formal process for clinical review and validation of the top-ranked hypotheses generated by the system, creating a tight feedback loop between automated discovery and expert-driven scientific inquiry.

Furthermore, we plan to establish a formal process for clinical review and validation of the top-ranked hypotheses generated by the system, creating a tight feedback loop between automated discovery and expert-driven scientific inquiry. As part of this we will also test the generalization of discovered features not only on held-out test sets, but across hospitals such as the SHC Tri-Valley dataset to ensure robustness of extracted knowledge between hospitals.

## 5 Conclusion

This paper introduces a fully automated system that leverages mechanistic interpretability to extract novel scientific hypotheses from the latent representations of medical foundation models. By training Matryoshka Sparse Autoencoders, we decompose complex features into an interpretable basis of clinical concepts. An automated pipeline then interprets these concepts and quantifies their prognostic value, culminating in the generation of scientific reports such as the one presented in this paper. We demonstrated the power of this approach through the proposal of an imaging biomarker for skin cancer that is more predictive at Stanford SHC than commonly collected clinical risk factors. This work represents a significant step towards transforming artificial intelligence from a tool for prediction into a collaborative partner in the process of scientific discovery, capable of systematically generating and prioritizing testable hypotheses at a scale previously unattainable.

**AI-agent Setup**    MechSci is a fully automated discovery process built on a multi-stage, agentic pipeline. A user simply provides a dataset and a set of clinical outcomes of interest, and the system runs end-to-end to discover interpretable hypotheses. A ready-to-use implementation of this agent will be made publicly available at https://github.com/RobbieHolland/MechSci. The core of the discovery process leverages Mechanistic Interpretability via SAEs (as described in Section 2.3) to discover interpretable features within the input data. MechSci then invokes an LLM (Gemini 2.5 Pro) to run automated concept interpretation (Section 2.4) and single-shot manuscript generation

(Section 2.5). When a user begins to explore generated hypotheses in the "Feature Discoverer" web application (Figure 2), MechSci gathers all pre-computed assets (statistics, plots, interpretations) and populates a prompt template for the LLM to produce the complete, submission-ready LaTeX manuscript for review or manual edits by the radiologist or clinical researcher.



## Responsible AI Statement

This work adheres to the NeurIPS Code of Ethics. The system was designed to ensure safe deployment in research settings only, with safeguards to prevent patient-identifiable information from being used. Broader impacts, such as reliance on AI-generated hypotheses, have been considered, and final interpretations remain under human expert supervision.

## Reproducibility Statement

All code for SAE training, LLM-based interpretation, and hypothesis generation will be made available (but would currently break anonymity for double-blind review). The study uses a public dataset from Merlin and includes detailed hyperparameters, evaluation protocols, and random seeds to ensure reproducibility. Figures and tables can be regenerated following the documented pipeline. We also provide the full prompt used to generate the article.

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

# A  Appendix

Below in A.1 is the prompt used to generate this paper in a single shot, subject to a small number of minor improvements prior to submission. Placeholders inside { } braces are replaced with specific statistics and values from the selected feature disease combination webpage. We also include the AutoInterp reasoning trace in A.2.

## A.1  Prompt used to generate paper in one shot

Your task is to write the main body of a LaTex paper to be submitted to the Agents4Science 2025 conference at Stanford. It is structured as an up-to-8 page paper. Write it like an expert academic, and in passive voice. Be comprehensive in writing the paper, the methodology and specific details are what makes this a strong paper. We're looking to get excellent reviews from NeurIPS-style reviewers.

Below I provide you information on the paper you are writing, including links to figures you created, and then you will think about how to write the paper before writing it out in full in one shot. Also, add citations provided using their numbers.

—- Our overall project is structured as follows. This will help you write the Abstract, Introduction and Methods: Uncovering Scientific Knowledge from Multimodal Medical Data through Foundation Models and Mechanistic Interpretability

Background and Motivation Modern healthcare generates a vast and complex array of multimodal data — CT imaging studies, radiology reports, laboratory results, clinical notes, and structured EHR data — all of which contain latent information about disease mechanisms and patient outcomes. Foundation models trained across these modalities have achieved exceptional diagnostic and prognostic performance, uncovering subtle relationships that elude conventional statistical approaches.

However, despite their accuracy, these models remain difficult to interpret. Techniques such as saliency maps or gradient-based attributions offer limited insights into what the models actually learn, particularly when applied across multiple data types. The challenge is not only to explain predictions but to expose the internal representations and mechanisms that connect to real biomedical phenomena.

This work explores a path toward mechanistic interpretability: understanding and restructuring the internal feature spaces of large multimodal foundation models through sparse autoencoders (SAEs). The goal is to convert opaque, high-dimensional representations into sparse, interpretable concept spaces that make model reasoning visible and scientifically meaningful.

Concept and Approach Foundation models capture intricate, multidimensional patterns that encode relationships across imaging, clinical text, and structured health data. These representations are powerful but dense and difficult to parse. Sparse autoencoders provide a systematic way to restructure these representations: by enforcing sparsity, they isolate the key activations that explain most of the variance in the data. Each sparse feature can then correspond to a clinically meaningful concept — for example, a specific imaging phenotype or laboratory pattern.

Prior work outside medicine has shown that such sparse representations can reveal interpretable internal concepts hidden within large models. Translating this approach to the medical domain can provide a way to map model features to physiological and pathological mechanisms. The hypothesis is that SAEs trained on foundation model embeddings will preserve predictive performance while exposing interpretable and clinically relevant structure.

Overview of Method This project leverages a large, IRB-approved multimodal dataset from Stanford Health Care (SHC) and Stanford Health Care Tri-Valley (SHC-TV), which includes CT scans, radiology reports, clinical documentation, laboratory results, ICD codes, and longitudinal outcome data for more than 45,000 patients.

Multimodal Representations Foundation model embeddings will be extracted from both text and imaging data. Large language models such as Gemini and Qwen will be used to embed clinical reports and notes, while Merlin, a multimodal medical foundation model developed by our group, will be used to obtain CT-based image embeddings. These representations will form the input space for subsequent sparse autoencoder training.

Sparse Autoencoder Training Sparse autoencoders will be trained to reorganize the dense foundation model features into sparse, interpretable concept spaces. We will implement and compare architectures such as Top-K SAEs and Matryoshka SAEs, evaluating how effectively they produce disentangled, meaningful concepts that preserve the informational richness of the original embeddings.

Automated Concept Interpretation To assign meaning to each sparse feature, we will construct an automated interpretability pipeline. For every SAE-derived concept, we will identify samples that strongly activate it and use large language or vision–language models to describe shared attributes and propose semantic labels. These candidate concept labels will be iteratively refined and validated on held-out samples to ensure reliability and clinical coherence.

Performance and Utility Assessment The resulting sparse concept representations will be evaluated for their ability to support clinically relevant predictive tasks. We will compare diagnostic and prognostic performance between the sparse concepts and the original foundation model embeddings, and assess whether accurate predictions can be achieved using a smaller, more interpretable subset of sparse features.

Future Directions Once validated, this framework will serve as a foundation for deeper scientific discovery. Sparse, interpretable features derived from foundation models could be used to explore mechanistic hypotheses about disease progression, treatment response, and outcome prediction. For example, newly identified latent concepts may correspond to previously unrecognized imaging biomarkers or physiological signatures that warrant clinical study.

Future work will expand to additional modalities such as genomics and longitudinal EHR data, and integrate causal inference tools to test whether identified concepts represent mechanistic pathways rather than statistical correlations. Further development will also focus on creating clinician-facing interfaces for interactive exploration of the sparse concept space, supporting hypothesis generation directly from model-derived insights.

In the long term, this approach aims to reposition AI in medicine — from systems that merely predict outcomes to systems that reveal structure, generate hypotheses, and advance understanding of human health and disease. By combining large-scale multimodal data from institutions like Stanford Health Care with mechanistic interpretability methods, we move toward an era where foundation models contribute not only to clinical decision-making but to genuine biomedical discovery.

References 1. Cunningham, H., et al. Sparse Autoencoders Find Highly Interpretable Features in Language Models. 2023. arXiv:2309.08600 DOI: 10.48550/arXiv.2309.08600. 2. Van Veen, D., et al., Adapted large language models can outperform medical experts in clinical text summarization. Nat Med, 2024. 30(4): p. 1134-1142. 3. Blankemeier, L., et al., Merlin: A Vision Language Foundation Model for 3D Computed Tomography. Res Sq, 2024. 4. Chen, Z., et al., Chexagent: Towards a foundation model for chest x-ray interpretation. arXiv preprint arXiv:2401.12208, 2024. 5. Gao, L., et al. Scaling and evaluating sparse autoencoders. 2024. arXiv:2406.04093 DOI:10.48550/arXiv.2406.04093. 6. Bussmann, B., et al. Learning Multi-Level Features with Matryoshka Sparse Autoencoders. 2025. arXiv:2503.17547 DOI: 10.48550/arXiv.2503.17547. 7. Templeton, A., et al. Scaling Monosemanticity: Extracting Interpretable Features from Claude 3 Sonnet. 2024.

—- Method specifics: Specifically, for this first iteration of the project, we use 25,334 CT scans and train four different Top-K SAEs using sparsity K=5, 10, 20 and 40. We use N=8192 dictionary size. This is a Matryoshka Top-K SAE which uses nested dictionary sizes of [128, 512, 2048 and 8192].

For AutoInterpretation we use LLMs, specifically Gemini 2.5 Pro using the same API. We provide them the anonymized radiology report findings, which describe the CT images in text by human experts. In particular, we provide 20 samples which highly activate the feature to the model, and ask it to summarize the common properties of these samples. Then, we test this interpretation on a held-out set of 20 other highly activating samples. This results in a percentage of generalization of the interpretation.

—- Results specifics: I have created a website which lists all the discovered SAE features and links them to prognostic risk for disease. We are currently on one of the subpages which links feature {feature_name} to disease {disease}. In total, given all SAE alive features across configurations, and their combinations with each disease, we have generated {number_of_hypotheses} hypotheses,

and therefore {number_of_hypotheses} LaTex papers such as this one. For this submission we have selected the generated paper for the hypothesis that feature {feature_name} is highly predictive of onset to {disease} within 0.5-7.5 years. Below we list arrays of statistics that back this up:

{statistics}

This feature was AutoInterpreted, and was given the following interpretation:

{interpretation}

We also compare to the best risk factors that have been discovered through human efforts. These are listed below. Please provide a comparison to each of these in the paper:

{best_established_risk_factors}

This approach, that we can automatically generate a LaTex paper for every SAE feature x disease combination, is also part of the Methods of this paper. On the website, there is a button I just pressed called "Generate LaTex paper" which led to this prompt being created, and then passed to the Gemini 2.5 Pro API. Describe in the paper how we autofill these statistics and figures straight from the website. Also comment how the generation of this entire paper is done in a single shot (make this clear from the abstract).

For the results, you should create tables where necessary based on the provided statistics. We also have several figures: - 'figures/sae_r2.jpg': Original SAE fit statistics, with $R\hat{2}$ on the original feature space reconstruction vs. the sparsity parameter K and number of alive features. The plot shows Matryoshka SAEs in orange and standard Top-K SAEs in blue. It shows a strong pareto improvement of the Matryoshka SAEs over the Top-K SAEs. - 'figures/sae_downstream.jpg': SAE downstream performance, which takes each SAE and uses logistic regressions to predict the binary onset of conditions. - 'figures/feature_discoverer_webapp.jpg': Shows a screenshot of the webpage for feature_name linked to disease, with the 'Generate LaTex paper' button circled in red. - 'figures/feature_disease_plot.jpg': Shows a histogram of feature_names activation bins on the xaxis, and the proportion of individuals in each bin which develop disease. - 'figures/best_odds_ratio_per_sae.jpg': 'Shows the best highest odds ratio feature yielded per SAE configuration for disease.' - 'figures/comparisons_to_human_features.jpg': Shows a barplot of odds ratios for the best established human risk factors (see the statistics for a list) and feature_name for progression to disease.

You may group figures together if you deem it helpful.

—- Discussion and Conclusion sections: Firstly, draw your own conclusions from the statistics provided for this feature and its comparison to existing risk factors. Write a discussion which summarize the strengths of this approach, the fully automatic nature of it, and the excitement around what it can discover.

Also within the discussion, you should mention the limitations of the approach, and how it could be improved, and future work based on the 'overall project structure' outlined in the beginning.

The following conclusion section should be a shorter, punchier version of that.

—- Formatting: For describing the feature and disease names, put '´before any '_' and wrap it in texttt. Finally the prompt used to generate this paper and AutoInterp reasoning trace are listed in the Appendix under ref{appendix:paper_gen_prompt} and ref{appendix:llm_autointerp_reasoning}.

## A.2 LLM AutoInterp reasoning

This feature was AutoInterpreted, and was given the following interpretation:

My reasoning process to determine the neuron's detected feature is as follows:

1. **Initial Scan of Reports for Overarching Themes:** I began by reading through all 15 reports to identify any high-level, recurring concepts. Two themes immediately stood out: (1) evidence of prior surgery or current medical intervention, and (2) signs of active inflammation or malignancy. * **Surgery/Intervention:** This was evident from frequent mentions of "surgically absent" organs (especially the gallbladder), "postsurgical changes," "transplant," and the presence of numerous medical devices like catheters, drains, stents, and LVADs. Reports 1, 3, 4, 6, 7, 8, 10, 12, 13, and 14 are strong examples of this. * **Inflammation/Malignancy:** This theme was clear from

descriptions of "inflammation," "peritonitis," "diverticulitis," "phlegmon," "necrotic lymph nodes," and "infiltrative soft tissue." Reports 2, 4, 6, 8, 11, and 15 are strong examples.

2. **Identifying the Common Visual Denominator:** A neuron detects visual patterns, not abstract medical concepts. My next step was to determine the common visual appearance that underlies both "surgery" and "inflammation" on a CT scan. * Surgery inherently disrupts normal anatomy. Incisions create scars, resections leave altered tissue planes, and post-operative healing often involves fluid collections (seromas, hematomas) and inflammatory reactions in the surrounding tissues. * Inflammation (like diverticulitis or peritonitis) and infiltration by tumors also disrupt normal anatomy. The key visual sign for this on CT is often "fat stranding"—a hazy, ill-defined increase in the density of the normally black abdominal fat—as well as fluid collections and abnormal soft tissue masses.

3. **Hypothesis Formulation and Testing:** I hypothesized that the core feature is the visual manifestation of this disruption. I considered several specific phrasings: * *Hypothesis A: "Indwelling medical devices."* This was strong for many reports (e.g., 3, 14) but failed to explain the high activation for reports describing only inflammation (15) or a necrotic mass (2). * *Hypothesis B: "Abnormal fluid collections."* This was present in many cases but was explicitly absent in several high-ranking reports, including the 1 report ("no free fluid"). * *Hypothesis C: "Disruption of abdominal fascial planes."* This was a strong conceptual candidate. Both surgery and inflammation blur the sharp lines that normally define anatomical structures. This concept successfully explained all 15 cases. For example, a surgical incision (1), an infiltrative mass (2), a hematoma with stranding (3), peritonitis (4), and diverticulitis (15) all visually disrupt or blur fascial planes.

4. **Refining the Description for Precision:** The concept of "disruption of fascial planes" is correct, but I needed to translate it into a more specific, descriptive radiological term that captures the primary visual evidence. The most consistent visual signs of this disruption across the reports are abnormal densities within the normally dark fat of the abdomen. This includes the hazy pattern of "stranding" and the more solid appearance of "soft tissue density" (from fluid, phlegmon, scar, or tumor). Therefore, I combined these observations into a single, comprehensive feature description.

5. **Final Verification:** I re-checked my final proposed feature—"Abnormal soft tissue density and stranding within mesenteric, peritoneal, or retroperitoneal fat"—against the highest-ranking reports. * Report 1: "post surgical changes" and "surgical clips" represent abnormal densities in the abdominal wall and retroperitoneal fat. * Report 2: "necrotic ... lymph nodes and soft tissue implants" are abnormal soft tissue densities in the pelvic/retroperitoneal fat. * Report 3: "hematoma" and "stranding" are a perfect match. * Report 4: "peritonitis" and "fluid collection" match the description. * Report 15 (an inflammatory example): "pericolic fat stranding" is a direct match. This description successfully unifies the varied findings (postsurgical changes, inflammation, fluid collections, and masses) into a single, coherent visual feature that a neuron could plausibly detect. It adheres to the prompt's constraints by being specific, avoiding "or," and focusing on appearance rather than pathology.

* This feature represents abnormal soft tissue density and stranding within mesenteric, peritoneal, or retroperitoneal fat.

