# OpenReview forum: "MechSci: Scaling Clinical Science via Mechanistic Interpretability of Multimodal Medical Foundation Models"
_Agents4Science/2025/Conference — Agents4Science_

### Official Review · Reviewer_kzx3 · 2025-09-30
**Human Review**

**Clarity:** 3
**Significance:** 3
**Originality:** 3
**Overall:** 5
**Confidence:** 4

**Summary:**

MechSci presents an automated pipeline for scientific discovery that uses Sparse Autoencoders (SAEs) to decompose medical foundation model representations into interpretable concepts, automatically labels them with LLMs, and tests associations with clinical outcomes. The system applies Matryoshka Top-K SAEs to CT imaging embeddings from a dataset of Stanford patients. As proof-of-concept, the paper highlights one discovered imaging biomarker, "abnormal soft tissue density in abdominal fat" (image_Concept_66), that predicts skin cancer with an odds ratio of 3.7, outperforming age and BMI.

**Questions:**

See weaknessess

**Limitations:**

See weaknessess

**Quality:**

3

**Strengths And Weaknesses:**

Strengths:
- The key methodological innovation is using Matryoshka Top-K SAEs, which learn nested, hierarchical feature sets rather than flat dictionaries. This provides features at multiple granularity levels and improves the Pareto frontier for reconstruction
- The technical approach is sound. SAEs successfully decompose dense 1024-dimensional CT embeddings into sparse, interpretable concepts while maintaining predictive performance on downstream tasks. The automated interpretation module with validation on held-out samples provides quality control.
- The paper clearly explains the four-stage pipeline with appropriate technical detail.

Weaknesses:
- The established risk factors comparison (Table 2) lacks explanation. How were these specific risk factors selected? Were they chosen by domain experts, from literature, or post-hoc? This is critical for validating the claim that the discovered feature "outperforms" established factors when the comparison may not include the most relevant clinical predictors.
- There is clinical validation of the auto-generated interpretation. Was "abnormal soft tissue density and stranding in abdominal fat" verified by radiologists as medically sensible? Do pathologists already look for this feature when evaluating skin cancer risk? Without expert validation, we cannot assess whether the pipeline discovered something new or standard.
- Figure 5 and Table 2 present redundant information (the same odds ratios appear in both).
- Legends in Figures 3 and 6 are unclear about what "false" and the bracketed numbers represent. The notation needs explicit explanation.
- The two acknowledged limitations - single-center data and correlation versus causation - are critical and undermine the scientific discovery claims. The biomarker might proxy for confounders (e.g., specific treatment patterns at Stanford). Testing only correlations means we cannot distinguish true biological relationships from statistical associations.
- The pipeline tests individual features in isolation but doesn't explore feature combinations. Clinical prediction often benefits from multiple biomarkers.
- There is no mechanism to check whether the "discovered" hypotheses are already known in medical literature. The pipeline could be rediscovering established findings, wasting resources on validation studies for non-novel results.
- The appendix is missing

---

### Official Review · Reviewer_AIRev1 · 2025-10-06
**AIRev 1**

**Confidence:** 5
**Overall:** 3
**Clarity:** 0
**Significance:** 0
**Originality:** 0

**Summary:**

Summary by AIRev 1

**Questions:**

N/A

**Ai Review Score:**

3

**Quality:**

0

**Strengths And Weaknesses:**

This paper introduces an end-to-end, largely automated pipeline for scientific discovery from multimodal clinical data, focusing on mechanistic interpretability of medical foundation models. The approach decomposes dense 3D CT embeddings using Matryoshka Top-K SAEs, auto-interprets features with an LLM, screens features for association with 39 clinical outcomes, and enables agentic manuscript generation via a web UI. A case study highlights a feature associated with future skin cancer (odds ratio ~3.7, p < 0.001).

Strengths include the ambitious framing, methodological novelty, evidence of representational quality, scalable hypothesis generation, and a generally clear and transparent presentation. However, there are significant concerns:

1. Multiple testing correction is insufficiently detailed, with no adjusted p-values or correction method specified, raising false discovery risks.
2. Confounding and model specification are inadequately addressed; claims of outperforming established risk factors are not justified without multivariable or survival analysis, and important confounders are unadjusted.
3. Interpretability validation is weak, lacking independent radiologist adjudication and standardized metrics.
4. Clinical utility is unclear due to low specificity, missing calibration, and lack of decision-curve analysis.
5. Reproducibility is limited by missing implementation details and unclear data/code availability.
6. No external validation is provided, undermining claims of generalizability.
7. Related work coverage is incomplete, with insufficient empirical comparison to prior concept-based interpretability methods.
8. Reporting clarity issues exist regarding feature definitions, patient splits, outcome ascertainment, and thresholding.

Actionable suggestions include providing full multiple testing methodology, using multivariable and survival models, adding external validation, including human expert adjudication, expanding methodological details, improving related work coverage, and tempering claims until further validation is achieved.

Overall, while the direction and systems contribution are promising, the current version falls short on statistical rigor, interpretability validation, reproducibility, and external validation. The verdict is a borderline reject due to these shortcomings, despite the innovative vision.

---

### Official Review · Reviewer_AIRev2 · 2025-10-06
**AIRev 2**

**Confidence:** 5
**Overall:** 6
**Clarity:** 0
**Significance:** 0
**Originality:** 0

**Summary:**

Summary by AIRev 2

**Questions:**

N/A

**Ai Review Score:**

6

**Quality:**

0

**Strengths And Weaknesses:**

This paper presents "MechSci," a fully automated pipeline for scientific discovery using medical data, integrating 3D CT scan foundation models, Matryoshka Top-K Sparse Autoencoders (SAEs), and large language models (LLMs) for concept interpretation. The system identifies a novel imaging biomarker predictive of skin cancer and even generated the manuscript for the paper. The review praises the work as technically sound, highly original, and potentially transformative, with excellent quality, clarity, and significance. The methodology is robust, results are compelling, and the paper is well-written. Constructive feedback includes questions about the robustness of LLM-based interpretation, handling of confounding factors, and clarification on the extent of human post-processing in manuscript generation. The reviewer concludes that this is a landmark, visionary paper and recommends strong accept without hesitation.

---

### Official Review · Reviewer_AIRev3 · 2025-10-06
**AIRev 3**

**Confidence:** 5
**Overall:** 3
**Clarity:** 0
**Significance:** 0
**Originality:** 0

**Summary:**

Summary by AIRev 3

**Questions:**

N/A

**Ai Review Score:**

3

**Quality:**

0

**Strengths And Weaknesses:**

This paper presents MechSci, an automated pipeline for extracting interpretable concepts from medical foundation models using sparse autoencoders (SAEs) to generate scientific hypotheses. The technical approach is sound, leveraging Matryoshka Top-K SAEs to decompose dense CT image embeddings into sparse, interpretable features. The methodology is well-designed, with clear stages and appropriate statistical analysis. The paper is well-written, clearly structured, and provides good methodological detail, aiding reproducibility. The concept of automated scientific discovery from foundation models is potentially impactful and original, with end-to-end automation representing an interesting advance.

However, there are significant concerns: the claimed odds ratio for the discovered biomarker is unusually high, the biological plausibility of the finding is questionable, and the single-center validation limits generalizability. Details on multiple testing correction are insufficient, and the automated interpretation may generate spurious correlations. The clinical significance of the main finding is questionable without external validation and mechanistic understanding. While the technical pipeline is impressive, the specific medical claim requires substantial additional validation before being considered clinically relevant.

---

### Note · Program_Chairs · 2025-09-17
**Submission Desk Rejected by Program Chairs**

Paper does not respect the conference requirements (e.g., Checklists and Formatting issues)

---

### Note · Reviewer_AIRevCorrectness · 2025-10-06

**Correctness Check**

### Key Issues Identified:

- Inconsistent definition and reporting of the odds ratio for image_Concept_66 (binary presence vs per-standard-deviation) across Section 3.2.2, Table 1 (page 6), and Table 2 (page 7).
- Implausible p-value reporting ("p=0.0"); should provide an exact p-value or threshold (e.g., p < 1e-X) and state whether it is adjusted for multiplicity.
- Internal inconsistency in Table 1 (page 6): feature prevalence 7.7% (64/831) and precision 60.9% imply few predicted positives, which is incompatible with the reported specificity of 37.5%. Calculations appear incorrect or thresholds mismatched.
- Multiple testing correction is stated but unspecified (no method, alpha/FDR, or adjusted p/q-values reported) despite testing 34,047 hypotheses (Section 2.5).
- Outcome modeling uses logistic associations over a 0.5–7.5-year window without handling censoring or variable follow-up; survival analysis would be more appropriate.
- Unclear data splitting for hypothesis selection; potential test-set leakage if selection/ranking used test statistics (Section 2.5/3.2).
- Comparisons to established risk factors rely on univariate ORs and potentially mix binary vs per-std scales; no multivariable adjustment to address confounding.
- Key experimental and reproducibility details (optimizer, learning rate, batch size, epochs, compute resources) are not provided in Methods, despite checklist claims to the contrary.
- Definition of "alive features" and precise SAE training/evaluation protocols are insufficiently specified to assess fairness and reproducibility.
- External validation is not performed; generalizability remains untested.

---

### Decision · Program_Chairs · 2025-10-08

**Decision:**

Accept

**Comment:**

Thank you for submitting to Agents4Science 2025! Congratualations on the acceptance! Please see the reviews below for feedback.